# Single-Pass Pivot Algorithm for Correlation Clustering. Keep it simple!

**Sayak Chakrabarty**
Northwestern University

**Konstantin Makarychev**[*]
Northwestern University

## Abstract

We show that a simple single-pass semi-streaming variant of the Pivot algorithm for Correlation Clustering gives a $(3 + \varepsilon)$-approximation using $O(n/\varepsilon)$ words of memory. This is a slight improvement over the recent results of Cambus, Kuhn, Lindy, Pai, and Uitto, who gave a $(3 + \varepsilon)$-approximation using $O(n \log n)$ words of memory, and Behnezhad, Charikar, Ma, and Tan, who gave a 5-approximation using $O(n)$ words of memory. One of the main contributions of this paper is that both the algorithm and its analysis are very simple, and also the algorithm is easy to implement.

## 1 Introduction

In this paper, we provide a simple single-pass streaming algorithm for Correlation Clustering. The Correlation Clustering model was introduced by Bansal, Blum, and Chawla [2004]. In this model, we have a set of objects represented by vertices of a graph. A noisy binary classifier labels all edges of the graph with "+" and "-" signs. We call edges labeled with a "+" sign – positive edges and edges labeled with a "-" sign – negative edges. A positive edge indicates that the endpoints of the edge are similar, and a negative edge indicates that the endpoints are dissimilar. Given a labeled graph, our goal is to find a clustering that disagrees with the minimum number of edge labels. A positive edge disagrees with the clustering if its endpoints are in different clusters, and a negative edge disagrees with the clustering if its endpoints are in the same cluster. We assume that the classifier can make many errors and, consequently, there may be no clustering that agrees with all labels of the graph.

Correlation Clustering has been widely studied in ML and theoretical computer science communities. It was used for image-segmentation (Kim et al. [2014]), link prediction (Yaroslavtsev and Vadapalli [2018]), document clustering (Bansal et al. [2004]), community detection (Veldt et al. [2018], Shi et al. [2021]), and other problems.

In Section 1.1, we discuss various objectives and assumptions used for Correlation Clustering. We now focus on the most common setting, where the graph $G$ is a complete graph (i.e., we have a "+" or "-" label for each pair of vertices $u$ and $v$), the classifier can make arbitrary mistakes, and the objective is to minimize the number of disagreements.

In their original paper, Bansal, Blum, and Chawla [2004] gave a constant factor approximation for Correlation Clustering on complete graphs. Charikar, Guruswami, and Wirth [2005] improved the approximation factor to 4. Then, Ailon, Charikar, and Newman [2008] proposed combinatorial and LP-based PIVOT algorithms, with approximation factors 3 and 2.5, respectively. The LP-based PIVOT algorithm was further improved by Chawla, Makarychev, Schramm, and Yaroslavtsev [2015], who provided a 2.06 approximation. Finally, Cohen-Addad, Lee, and Newman [2022b] and Cohen-Addad, Lee, Li, and Newman [2023] used a Sherali-Adams relaxation for the problem to obtain $1.994 + \varepsilon$ and $1.73 + \varepsilon$ approximations, respectively. On the hardness side, Charikar, Guruswami, and Wirth [2005] proved that the problem is APX-Hard and thus unlikely to admit a PTAS.

---

[*]Corresponding author.

37th Conference on Neural Information Processing Systems (NeurIPS 2023).

Most abovementioned algorithms use linear programming (LP) for finding a clustering. The number of constraints in the standard LP is $\Theta(n^3)$ (due to "triangle inequality constraints"). Solving such LPs is prohibitively expensive for large data sets[2]. That is why, in practice, the algorithm of choice is the combinatorial PIVOT algorithm by Ailon et al. [2008] along with its variants. The approximation factor of this algorithm is 3.

To deal with massive data sets, researchers started exploring parallel (MPC) and semi-streaming algorithms. Recent works on MPC algorithms include Blelloch, Fineman, and Shun [2012], Chierichetti, Dalvi, and Kumar [2014], Pan, Papailiopoulos, Oymak, Recht, Ramchandran, and Jordan [2015], Ahn, Cormode, Guha, McGregor, and Wirth [2015], Ghaffari, Gouleakis, Konrad, Mitrović, and Rubinfeld [2018], Fischer and Noever [2019], Cambus, Choo, Miikonen, and Uitto [2021], Cohen-Addad, Lattanzi, Mitrović, Norouzi-Fard, Parotsidis, and Tarnawski [2021], Assadi and Wang [2022], Cambus, Kuhn, Lindy, Pai, and Uitto [2022], Behnezhad, Charikar, Ma, and Tan [2022].

Streaming algorithms have been extensively studied by Chierichetti, Dalvi, and Kumar [2014], Ahn, Cormode, Guha, McGregor, and Wirth [2015], Ghaffari, Gouleakis, Konrad, Mitrović, and Rubinfeld [2018], Cohen-Addad, Lattanzi, Mitrović, Norouzi-Fard, Parotsidis, and Tarnawski [2021], Assadi and Wang [2022], Cambus, Kuhn, Lindy, Pai, and Uitto [2022], Behnezhad, Charikar, Ma, and Tan [2022, 2023]. This year, Behnezhad, Charikar, Ma, and Tan [2023] designed a single-pass polynomial-time semi-streaming 5-approximation algorithm that uses $O(n)$ words of memory. Then, Cambus, Kuhn, Lindy, Pai, and Uitto [2022] provided a single-pass polynomial-time semi-streaming $(3 + \varepsilon)$-approximation algorithm that uses $O(n \log n)$ words of memory. Both algorithms are based on the combinatorial PIVOT algorithm.

**Our Results.** In this paper, we examine a simple semi-streaming variant of the Ailon, Charikar, and Newman [2008] combinatorial PIVOT algorithm that uses $O(n/\varepsilon)$ words of memory and gives a $3 + \varepsilon$ approximation. Our algorithm needs less memory than the algorithm by Cambus, Kuhn, Lindy, Pai, and Uitto [2022] and gives the same $(3 + \varepsilon)$ approximation guarantee. The algorithm is very simple and easy to implement. Its analysis is simple and concise.

We note that one can take advantage of the low memory footprint of our algorithm even in a non-streaming setting. We provide some experimental results in Appendix (see Figure 2).

**Semi-streaming Model.** We consider the following single-pass semi-streaming model formally defined by Feigenbaum, Kannan, McGregor, Suri, and Zhang [2005] (see also Muthukrishnan [2005]). The algorithm gets edges of the graph along with their labels from an input stream. The order of edges in the graph is arbitrary and can be adversarial. The algorithm can read the stream only once. We assume that the algorithm has $O(kn)$ words of memory, where $k$ is a constant. Thus, it cannot store all edges in the main memory. As in many previous papers on streaming algorithms for Correlation Clustering, we shall assume that the stream contains only positive edges (if negative edges are present in the stream, our algorithm will simply ignore them). We prove the following theorem.

**Theorem 1.1.** *Algorithm 1 (see Figure 1) is a randomized polynomial-time semi-streaming algorithm with an approximation factor $3 + O(1/k)$. The algorithm uses $O(kn)$ words of memory, where each word can store numbers between $1$ and $n$. The algorithm spends at most $O(\log k)$ units of time for every edge it reads in the streaming phase. Then, it requires additional $O(kn)$ units of time to find a clustering (in the pivot selection and clustering phase).*

## 1.1 Related Work

In this paper, we consider the most commonly studied objective, MINDISAGREE, of minimizing the number of disagreements. Swamy [2004] and Charikar, Guruswami, and Wirth [2005] investigated a complementary objective, MAXAGREE, of maximizing the number of agreements. They gave 0.766-approximation algorithms for MAXAGREE. Puleo and Milenkovic [2016], Chierichetti, Kumar, Lattanzi, and Vassilvitskii [2017], Charikar, Gupta, and Schwartz [2017], Ahmadi, Khuller, and Saha [2019], Kalhan, Makarychev, and Zhou [2019], Ahmadian, Epasto, Kumar, and Mahdian [2020], Friggstad and Mousavi [2021], Jafarov, Kalhan, Makarychev, and Makarychev [2021], Schwartz and Zats [2022], Ahmadian and Negahbani [2023], Davies, Moseley, and Newman [2023] considered several variants of fair and local objectives.

---

[2]Veldt [2022] recently proposed a practical linear programming algorithm for Correlation Clustering.

**Initialization phase**
- Pick a random ordering of vertices $\pi : V \to \{1, \ldots, n\}$.
- For each vertex $u \in V$:
  - Create a priority queue $A(u)$ *capped at size $k$* and initialize $A(u) = \{u\}$.

**Streaming phase**
- For each edge $(u, v) \in E$:
  - Add $u$ to $A(v)$. Add $v$ to $A(u)$. Remove the lowest ranked elements from $A(u)$ and $A(v)$, if necessary, to make sure that they contain at most $k$ elements.

**Pivot selection and clustering**
- For each not-yet clustered vertex $u \in V$ chosen in the order of $\pi$:
  - Find the highest ranked vertex $v$ in $A(u)$ such that $v = u$ or $v$ is a pivot i.e.,

$$v = \operatorname*{argmin}_{v \in A(u)} \{\pi(v) : v = u \text{ or } v \text{ is a pivot}\}.$$

  - If such vertex $v$ exists, place $u$ in the cluster of $v$. If $u = v$, mark $u$ as a *pivot*.
  - Else, place $u$ in a singleton cluster. Mark $u$ as a *special singleton*.
- For each pivot vertex $u$, output the cluster consisting of all vertices clustered with $u$. Then, output all special singleton vertices, each in its own clusters.

Figure 1: Semi-Streaming Algorithm for Correlation Clustering

Charikar, Guruswami, and Wirth [2005] and Demaine, Emanuel, Fiat, and Immorlica [2006] provided $O(\log n)$ approximation for Correlation Clustering on incomplete graphs and edge-weighed graphs. For the case when all edge weights are in the range $[\alpha, 1]$, Jafarov, Kalhan, Makarychev, and Makarychev [2020] gave a $(3 + 2\ln{^1/_\alpha})$-approximation algorithm.

The problem has also been studied under the assumption that the classifier makes random or semi-random errors by Bansal, Blum, and Chawla [2004], Elsner and Schudy [2009], Mathieu and Schudy [2010], Makarychev, Makarychev, and Vijayaraghavan [2015]. Online variants of Correlation Clustering were considered by Mathieu, Sankur, and Schudy [2010], Lattanzi, Moseley, Vassilvitskii, Wang, and Zhou [2021], Cohen-Addad, Lattanzi, Maggiori, and Parotsidis [2022a].

## 2 Single-Pass Streaming Algorithm

Our algorithm is an extension of the algorithm by Behnezhad, Charikar, Ma, and Tan [2023]. Loosely speaking, the algorithm works as follows: It picks a random ranking of vertices, scans the input stream and keeps only the $k$ top-ranked neighbours for every vertex $u$. Then, it runs the Ailon, Charikar, and Newman [2008] PIVOT algorithm on the graph that contains only edges from every vertex to its $k$ top-ranked neighbours. We provide the details below.

We shall assume that every vertex is connected with itself by a positive edge. In other words, we include each vertex in the set of its own (positive) neighbours. We denote the set of positive neighbours of $u$ by $N(u)$. The algorithm first picks a random ordering of vertices $\pi : V \to \{1, \ldots, n\}$. We say that vertex $u$ is ranked higher than vertex $v$ according to $\pi$ if $\pi(u) < \pi(v)$. Therefore, $\pi^{-1}(1)$ is the highest ranked vertex, and $\pi^{-1}(n)$ is the lowest ranked vertex. For every vertex $u$, we will keep track of its $k$ highest-ranked neighbours. We denote this set by $A(u)$. We initialize each $A(u) = \{u\}$. Then, for every positive edge $(u, v)$ in the input stream, we add $u$ to the set of $v$'s neighbours $A(v)$ and $v$ to the set of $u$'s neighbours $A(u)$. If $A(u)$ or $A(v)$ get larger than $k$, we remove the lowest ranked vertices from these sets to make sure that the sizes of $A(u)$ and $A(v)$ are at most $k$. After we finish reading all edges from the input stream, we run the following variant of the PIVOT algorithm. We mark all vertices as non-pivots. Then, we consider vertices according to their ranking (i.e., $\pi^{-1}(1), \pi^{-1}(2), \ldots, \pi^{-1}(n)$). For every vertex $u$, we find the highest ranked neighbour $v$ in $A(u)$, such that $v = u$ or $v$ is a *pivot*. If such $v$ exists, we place $u$ in the cluster of $v$. Moreover, if $v = u$, we declare $u$ a *pivot*. If no such $v$ in $A(u)$ exists, we put $u$ in its own cluster and declare it a *special singleton*. We provide pseudo-code for this algorithm in Figure 1.

We use fixed-size priority queues to store sets $A(u)$. Whenever we need to add a vertex $v$ to $A(u)$, we insert $v$ into $A(u)$ and remove the lowest-ranked element from $A(u)$ if the size of $A(u)$ is greater

than $k$. We remark that even though each $u$ is always added to set $A(u)$ in the beginning of the algorithm, it may later be removed if $u$ is not among its $k$ top-neighbours.

We can reduce the memory usage of the algorithm by adding $u$ to $A(v)$ only if $\pi(u) < \pi(v)$ and adding $v$ to $A(u)$ only if $\pi(v) < \pi(u)$. This change does not affect the resulting clustering because $u$ is always added to $A(u)$, and, consequently, we never pick a pivot $v$ with $\pi(v) > \pi(u)$ at the clustering phase of the algorithm. Nevertheless, to simplify the exposition, we will assume that we always add $u$ to $A(v)$ and $v$ to $A(u)$.

In this paper, we will call clusters special singleton clusters if they were created in the "else" clause of the clustering phase. That is, if they contain a single vertex $u$, which is not a pivot. Note that some other clusters may contain a single vertex as well. We will not call them special singleton clusters.

Before analyzing the approximation factor of the algorithm, we briefly discuss its memory usage and running time. For every vertex $u$, we keep its rank $\pi(u)$, a set of its $k$ highest ranked neighbors $A(u)$, a pointer to the pivot if $u$ is not a special singleton, and two bits indicating whether $u$ is a pivot and whether $u$ is a special singleton cluster. Thus, the total memory usage is $O(kn)$ words; each word is an integer that can store a number between 1 and $n$.

The running time of the initialization step is $O(n)$. The worst-case running time of the streaming phase is $O(m \log k)$ where $m$ is the number of positive edges (the algorithm needs to perform a constant number of priority queue operations for each edge in the stream; the cost of each operation is $O(\log k)$). The *expected* running time of the streaming phase of the algorithm is[3] $O(m + nk \log k \log(2m/(nk)))$ if $m > nk$. It is $O(m)$ if $m > nk \log k \log \log k$. The running time of the clustering phase is $O(kn)$, since we need to examine all neighbors $v$ of $u$ in queue $A(u)$.

## 2.1 Approximation Factor

We now show that the approximation factor of the algorithm is $3 + O(1/k)$. For the sake of analysis, we consider an equivalent variant of our algorithm that reveals the ordering $\pi$ one step at a time. This variant of the algorithm is not a streaming algorithm, per se. However, it produces exactly the same clustering as our single-pass streaming algorithm. Thus, it is sufficient to show that it has a $3 + O(1/k)$ approximation. The algorithm, which we will refer to as *Algorithm 2*, works as follows. Initially, it marks all vertices as unprocessed and not clustered. It also sets a counter $K_1(u) = 0$ for each $u$. We denote the set of all unprocessed vertices at step $t$ by $V_t$ and the set of all non-clustered vertices by $U_t$. Then, $V_1 = V$ and $U_1 = V$. At step $t$, the algorithm picks a random vertex $w_t$ in $V_t$. If $w_t$ is not yet clustered ($w_t \in U_t$), then $w_t$ is marked as a pivot, a new cluster is created for $w_t$, and all not yet clustered neighbours of $w_t$ including $w_t$ itself are added to the new cluster. If $w_t$ is already clustered, then the counter $K_t(v)$ is incremented by 1 for all not yet clustered neighbours $v$ of $w_t$ (i.e., $K_{t+1}(v) = K_t(v) + 1$). After that, all vertices $v$, whose counter $K_{t+1}(v)$ got equal to $k$ at this step, are put into special singleton clusters. We mark $w_t$ as processed (by letting $V_{t+1} = V_t \setminus \{w_t\}$) and mark all vertices clustered at this step as clustered (by removing them from $U_t$).

We show that this algorithm outputs the same clustering as Algorithm 1. To this end, define an ordering $\pi$ for Algorithm 2 as $\pi : w_t \mapsto t$, where $w_t$ is the vertex considered at step $t$ of Algorithm 2. Clearly, $\pi$ is a random (uniformly distributed) permutation. We prove the following lemma.

**Lemma 2.1.** *If Algorithm 1 and Algorithm 2 use the same ordering $\pi$, then they produce exactly the same clustering of $V$.*

*Proof.* We show that every vertex $u$ is assigned to the same pivot $v$ in both algorithms, or $u$ is put in a special singleton cluster by both algorithms assuming that Algorithms 1 and 2 use the same ordering $\pi$. The proof is by induction on the rank $\pi(u)$. Consider a vertex $u$ and assume that all vertices $v$ higher ranked than $u$ are clustered in the same way by Algorithms 1 and 2. Both algorithms examine

---

[3]The probability that the $i$-th neighbour of $u$ is among the top $k$ already considered neighbours of $u$ is $\min(k/i, 1)$. Thus, the probability that the $i$-th neighbour is inserted in the priority queue $A(u)$ is $\min(k/i, 1)$. The expected number of neighbours inserted in the queue for vertex $u$ is at most $k + O(k \log(1 + |N(u)|/k))$. Each insertion takes $O(\log k)$ units of time. Therefore, the total time spent on the insertion operations is upper bounded by (here we use Jensen's inequality for function $\log x$)

$$O\left( \log k \cdot \sum_{u \in V} \left( k + k \log \left( 1 + \frac{|N(u)|}{k} \right) \right) \right) \leq O\left( nk \log k \cdot \left( 1 + \log \left( 1 + \frac{2m}{kn} \right) \right) \right).$$

neighbours of $u$ in exactly the same order, $\pi$. Moreover, they only consider the top $k$ neighbours: Algorithm 1 does so because it stores neighbours of $u$ in a priority queue of size $k$; Algorithm 2 does so because it keeps track of the number, $K_t(u)$, of examined neighbours of $u$ and once that number equals $k$, it puts $u$ in a special singleton cluster. Both algorithms assign $u$ to the first found pivot vertex $v$ or to a special singleton cluster if no such $v$ exists among the top $k$ neighbours of $u$. Hence, both algorithms cluster $u$ in the same way. □

We now analyze Algorithm 2. We split all steps of Algorithm 2 into *pivot* and *singleton* steps. We say that step $t$ is a pivot step if vertex $w_t$ which was processed at this step was not clustered at the beginning of the step (i.e., $w_t \in U_t$). We say that $t$ is a singleton step, otherwise (i.e., $w_t \in V_t \setminus U_t$). Observe that all clusters created at pivot steps are not special singleton clusters (even though some of them may contain a single vertex; see above for the definition of special singleton clusters). In contrast, all clusters created at singleton steps are special singleton clusters. We will now show that the expected cost of clusters created at pivot steps is at most $3OPT$. Then, we will analyze singleton steps and prove that the cost of clusters created at those steps is at most $O(1/k)\,OPT$, in expectation (where $OPT$ is the cost of the optimal solution).

We say that a positive or negative edge $(u, v)$ is settled at step $t$ if both $u$ and $v$ were not clustered at the beginning of step $t$ but $u$, $v$, or both $u$ and $v$ were clustered at step $t$. Note that once an edge $(u, v)$ is settled, we can determine if $(u, v)$ is in agreement or disagreement with the clustering produced by the algorithm: For a positive edge $(u, v) \in E^+$ if both $u$ and $v$ belong to the newly formed cluster, then $(u, v)$ is in agreement with the clustering. For a negative edge $(u, v) \in E^-$ if both $u$ and $v$ belong to the newly formed cluster, then $(u, v)$ is in disagreement with the clustering. Similarly, if only one of the endpoints $u$ or $v$ belongs to the new cluster, then $(u, v)$ is in agreement with the clustering for a negative edge $(u, v) \in E^-$ and in disagreement with the clustering for a positive edge $(u, v) \in E^+$. In the latter case, we say that a positive edge $(u, v) \in E^+$ is *cut* by the algorithm, and we call $(u, v)$ a *cut edge*.

We say that the cost of a settled edge is $0$ if it agrees with the clustering and $1$ if it disagrees with the clustering. Let $P$ be the cost of all edges settled during pivot steps. Furthermore, let $P^+$ be the cost of all positive edges and $P^-$ be the cost of all negative edges settled during pivot steps. Then, $P = P^+ + P^-$. If $v$ is a singleton cluster, let $S(v)$ be the number of positive edges incident to $v$ the algorithm cut when it created this cluster. Note, that some edges incident to $v$ might have been cut before $v$ was put in a singleton cluster. We do not count these edges in $S(v)$.

Every edge which is in disagreement with the clustering is (1) a positive edge cut at a pivot step of the algorithm; (2) a negative edge joined at a pivot step of the algorithm; or (3) a positive edge cut at a singleton step of the algorithm. Hence, the cost of the algorithm equals:

$$ALG = P^+ + P^- + \sum_{v \in V} S(v).$$

**Lemma 2.2.** *We have* $\mathbf{E}[P^+ + P^-] \le 3OPT$.

**Remark:** This lemma implies that $\mathbf{E}[P^+] \le 3OPT$. We note that we could get a slightly stronger bound: $\mathbf{E}[P^+] \le 2OPT$, which would yield a slightly stronger upper bound on $\mathbf{E}[\sum_v S(v)]$ (see below).

*Proof.* Observe that pivot steps are identical to the steps of the PIVOT algorithm. That is, given that $w_t$ belongs to $U_t$, the conditional distribution of $w_t$ is uniform in $U_t$ and the cluster created at this step contains $w_t$ and all its positive neighbours. Let us fix an optimal solution. Let $O^*$ be the set of edges in the optimal solution that disagree with the classifier. Denote the cost of the optimal solution by $OPT = |O^*|$. Ailon, Charikar, and Newman [2008] proved that the expected cost of the solution produced by the PIVOT algorithm is upper bounded by $3OPT$. Furthermore, they showed that the expected cost of edges settled at step $t$ is upper bounded by $3\mathbf{E}[\Delta OPT_t]$, where $\Delta OPT_t$ is the number of edges from $O^*$ settled by PIVOT at step $t$ (see also Lemma 4 in Chawla et al. [2015]). Therefore, the expected total cost of all edges settled at pivot steps of our algorithm is upper bounded by $3\mathbf{E}[\sum_{t \text{ is pivot step}} \Delta OPT_t]$. The number of all edges in $O^*$ settled at pivot steps is upper bounded by the size of $O^*$, which equals $OPT$. Hence, $\mathbf{E}[P^+ + P^-] \le 3OPT$. □

We now bound $\mathbf{E}[\sum_v S(v)]$. Fix a vertex $v$. Let $N(v)$ be the set of all vertices connected with $v$ by a positive edge ("positive or similar neighbours"), and $|N(v)|$ be the number of positive neighbours of

$v$ including $v$ itself. We now define a new quantity $X_t(v)$. If $v$ is not in a singleton cluster at step $t$, then let $X_t(v)$ be the number of positive edges incident on $v$ cut in the first $t-1$ steps of the algorithm. Otherwise, let $X_t(v)$ be the number of positive edges incident on $v$ cut by the algorithm before $v$ was placed in the singleton cluster. The number of edges incident to $v$ cut by the algorithm equals $X_{n+1}(v) + S(v)$. Hence, the total number of cut positive edges equals $\frac{1}{2} \sum_{v \in V} (X_{n+1}(v) + S(v))$. It also equals $P^+ + \sum_{v \in V} S(v)$. Thus, $\sum_{v \in V} X_{n+1}(v) = 2P^+ + \sum_{v \in V} S(v)$ and

$$\mathbf{E}\Big[ \sum_{v \in V} X_{n+1}(v) \Big] = \mathbf{E}\Big[ 2P^+ + \sum_{v \in V} S(v) \Big] \leq 6OPT + \mathbf{E}\Big[ \sum_{v \in V} S(v) \Big].$$

We now show that $\mathbf{E}[S(v)] \leq 1/k \, \mathbf{E}[X_{n+1}(v)]$ for all $v$ and, therefore,

$$\mathbf{E}\Big[ \sum_{v \in V} S(v) \Big] \leq \frac{6OPT}{k-1},$$

and

$$\mathbf{E}[ALG] = \mathbf{E}\Big[ P^+ + P^- + \sum_{v \in V} S(v) \Big] \leq \Big( 3 + \frac{6}{k-1} \Big) OPT.$$

To finish the analysis of Algorithm 2, it remains to prove the following lemma.

**Lemma 2.3.** *We have* $\mathbf{E}[S(v)] \leq 1/k \, \mathbf{E}[X_{n+1}(v)]$.

*Proof.* Define the following potential function:

$$\Phi_t(v) = \mathbf{1}(v \notin U_t) \cdot X_t(v) - K_t(v) \cdot (|N(v)| - X_t(v)),$$

where $\mathbf{1}(v \notin U_t)$ is the indicator of event $\{v \notin U_t\}$. Let $\mathcal{F}_t$ be the state of the algorithm at the beginning of step $t$. We claim that $\Phi_t(u)$ is a submartingale i.e., $\mathbf{E}[\Phi_{t+1}(u) \mid \mathcal{F}_t] \geq \Phi_t(u)$.

**Claim 2.4.** $\Phi_t(v)$ *is a submartingale.*

*Proof.* Consider step $t$ of the algorithm. If $v$ is already clustered (i.e., $v \notin U_t$), then all edges incident on $v$ are settled and terms $\mathbf{1}(v \notin U_t)$, $K_t(v)$, and $X_t(v)$ no longer change. Hence, the potential function $\Phi_t(v)$ does not change as well.

Suppose that $v \in U_t$. If at step $t$, the algorithm picks $w_t$ not from the neighbourhood $N(v)$, then $v$ does not get clustered at this step and also $K_t(v)$ does not change. The value of $X_t(v)$ may increase as some neighbours of $v$ may get clustered. However, $X_t(v)$ cannot decrease. Therefore, $\Phi_{t+1}(v) \geq \Phi_t(v)$, and $\mathbf{E}\big[\Phi_{t+1}(v) - \Phi_t(v) \mid w_t \notin N(v); \mathcal{F}_t\big] \geq 0$.

Let us now assume that at step $t$, the algorithm chooses $w_t$ from the neighbourhood of $v$. Then, $v$ is clustered with conditional probability at least

$$\Pr\Big( w_t \in N(v) \cap U_t \mid w_t \in N(v); \mathcal{F}_t \Big) = \frac{|N(v) \cap U_t|}{|N(v) \cap V_t|},$$

because if the pivot is chosen in $N(v) \cap U_t$, then $v$ is put in the cluster of $w_t$. We have $|N(v) \cap U_t| = |N(v)| - X_t(v)$, since every clustered neighbour of $v$ contributes 1 to the number of cut edges $X_t(v)$, and the total number of $v$'s positive neighbours equals $|N(v)|$. Also, note that $|N(v) \cap V_t| \leq |N(v)|$. Hence,

$$\Pr\Big( w_t \in N(v) \cap U_t \mid w_t \in N(v); \mathcal{F}_t \Big) \geq \frac{|N(v)| - X_t(v)}{|N(v)|}.$$

With this conditional probability, $\mathbf{1}(v \notin U_t)$ gets equal to 1 at step $t$, and, as a result, $\Phi_t(v)$ increases by $X_{t+1}(v) \geq X_t(v)$. With the remaining probability, $K_t(v)$ increases by 1, and, as a result, $\Phi_t(v)$ decreases by at most $|N(v)| - X_{t+1}(v) \leq |N(v)| - X_t(v)$. Thus,

$$\mathbf{E}\big[\Phi_{t+1}(v) - \Phi_t(v) \mid w_t \in N(v); \mathcal{F}_t\big] \geq \frac{|N(v)| - X_t(v)}{|N(v)|} \cdot X_t(v) - \frac{X_t(v)}{|N(v)|} \cdot (|N(v)| - X_t(v)) = 0.$$

This concludes the proof. $\qquad\square$

Since $\Phi_1(v) = 0$ and $\Phi_t(v)$ is a submartingale, we have $\mathbf{E}[\Phi_{n+1}(v)] \geq 0$. Therefore,

$$\mathbf{E}[X_{n+1}(v)] = \mathbf{E}[\mathbf{1}(v \notin U_{n+1}) \cdot X_{n+1}(v)] \geq \mathbf{E}[K_{n+1}(v) \cdot (|N(v)| - X_{n+1}(v))].$$

If $v$ is in a singleton cluster, then $K_{n+1}(v) = k$ and $X_{n+1}(v) + S(v) = |N(v)|$ (because all positive edges incident to $v$ are cut). Hence,

$$kS(v) = K_{n+1}(v) \cdot (|N(v)| - X_{n+1}(v)).$$

If $v$ is not in a singleton cluster, then $S(v) = 0$. Thus, we always have $kS(v) \leq K_{n+1}(v) \cdot (|N(v)| - X_{n+1}(v))$. Consequently, $\mathbf{E}[X_{n+1}(v)] \geq k\mathbf{E}[S(v)]$. $\square$

## Acknowledgments and Disclosure of Funding

Konstantin Makarychev is supported by NSF Awards CCF-1955351, CCF-1934931, EECS-29 2216970. The authors would like to thank the anonymous reviewers for their valuable comments.

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

# A  Figures

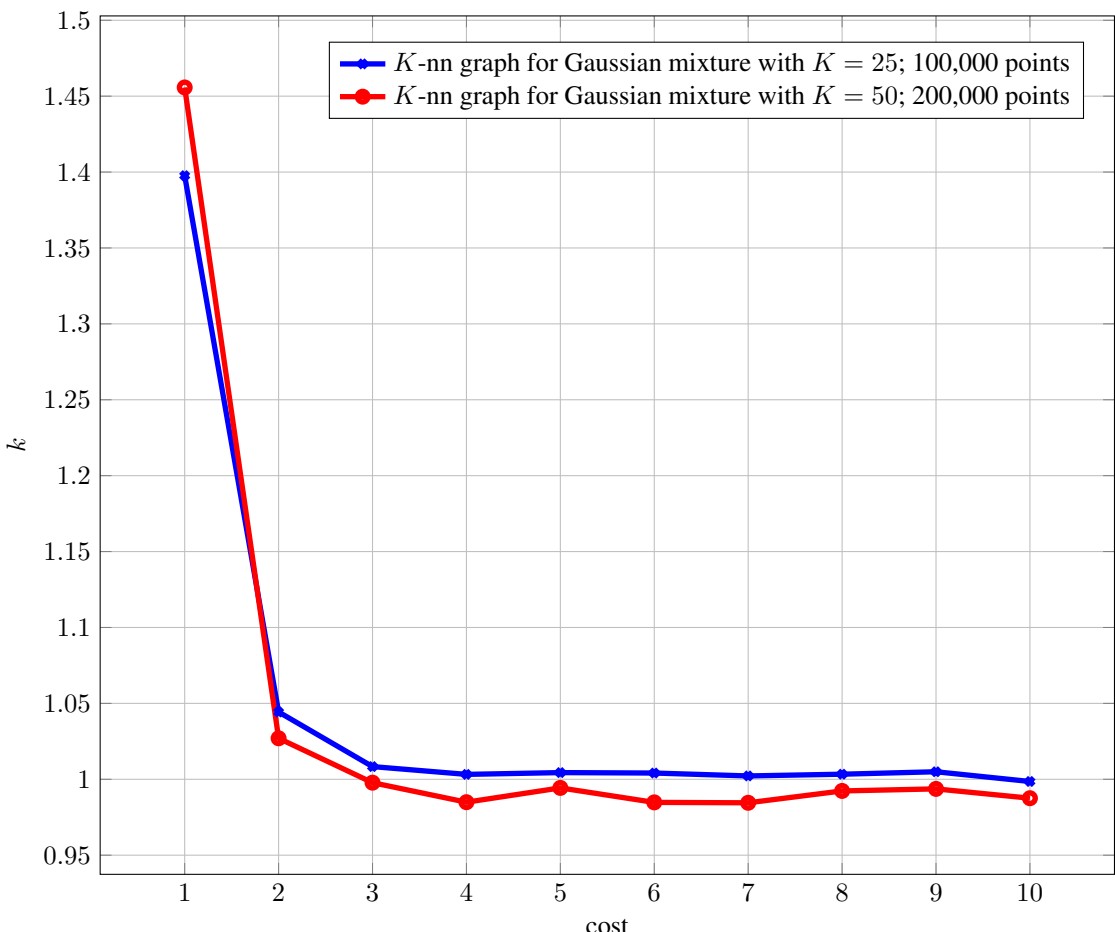

Figure 2: The cost of the clustering obtained by our algorithm comparing with the cost of the clustering obtained by the PIVOT algorithm for different values of $k$. The input graph is the $K$-nearest neighbour graph for the Gaussian mixture model.

