# OpenReview forum: "Single-Pass Pivot Algorithm for Correlation Clustering. Keep it simple!"
_NeurIPS.cc/2023/Conference — NeurIPS 2023 poster_

### Official Review · Reviewer_NEv2 · 2023-06-09

**Soundness:** 4 excellent
**Presentation:** 3 good
**Contribution:** 2 fair
**Rating:** 4
**Confidence:** 4

**Summary:**

The paper shows a single-pass semi streaming algorithm for correlation clustering (minimizing disagreements) which computes a  $3+\epsilon$ approximate solution using $O(n/\epsilon)$ words. This should be compared with:
 * a 5-approximate algorithm using O(n) words
 * a $3+\epsilon$ approximate algorithm using $O(n \log n)$ words

The algorithm is a variant of the PIVOT algorithm combined with a simple sparsification technique which keeps $1/\epsilon$ edges per each node.

**Strengths:**

* The paper improves the space usage for an important problem
* The presentation is clear. Actually the entire paper, including all proofs (excl references) fits in 6 pages

**Weaknesses:**

While I appreciate the theoretical result and the simplicity, the improvement over the previous results seems quite small: i.e. it can be seen as either slightly reducing the approximation factor or space per vertex from $O(\log n)$ to $O(1/\epsilon)$. I'm not convinced that the problem of correlation clustering in a streaming setting is popular enough for this kind of improvement to be of sufficient interest to NeurIPS audience

**Questions:**

I'm afraid I can't think of a question that could affect my recommendation

---

### Official Review · Reviewer_mSWo · 2023-07-01

**Soundness:** 4 excellent
**Presentation:** 4 excellent
**Contribution:** 3 good
**Rating:** 7
**Confidence:** 4

**Summary:**

The paper studies the correlation clustering problem in the semi-streaming model obtaining a $(3+\varepsilon)$-approximation to the problem with $O(\frac{n}{\varepsilon})$ words of memory. This is an improvement over [CKLPU'22] who gave the same approximation with $O(n\log n)$ words of memory (I'm not sure of the dependence on $\varepsilon$) and over [BCMT'22] who gave a 5-approximation with $O(n)$ words of memory. Moreover, the algorithm is extremely simple. It picks a a random ranking of the vertices and for each vertex keeps a list of its $k=O(1/\varepsilon)$ highest ranked neighbors during the streaming phase. In post processing the algorithm goes through the vertices according to their rank, for each vertex $u$ finding the highest ranked vertex $v$ in its list such that $v=u$ or $v$ is a pivot. In the former case, $u$ is declared a pivot, in the latter case, $u$ is assigned to the cluster of $v$. If no such $v$ exists, $u$ is put in a singleton cluster. The analysis describes an equivalent offline algorithm which is analysed partly via an analogue to the  Pivot algorithm from [ACN'08] and partly via a martingale argument for singleton vertices relating their cost to that of the Pivot algorithm.

**Strengths:**

The paper is very well written and the proofs are neat and readily checked to be correct. The algorithm is simple and I think it is nice and rare with a short paper with such a clean idea. Since furthermore, correlation clustering is of interest to many people in the Neurips community I think the paper should be accepted.

**Weaknesses:**

Perhaps the authors could have considered running some experiments on the quality of the clustering output by their algorihtm.

**Questions:**

What's the dependence on $\varepsilon$ in [CKLPU'22]?

l66: ".... has $O(n)$ words of memory where $k$ is a constant": Do you mean $O(nk)$?

l131: I think the runtime bound can be improved to $O((k\log k \log n)n+m)$ or something like that. Indeed, with the random order, you only need to update the priority queue with probability $k/i$ when the $i$'th neighbor arrives.

l142: You define $U_t$ to be the unclustered vertices but latter on in the martingale argument (l206, l209, l216,...) you refer to $U_t$ as the set of clustered vertices. I think the indicator in the potential should be $1(v\notin U_t)$.

l195-196: Can you define $\Delta OPT_t$ more clearly. When you say the optimal cost of the edges settled by PIVOT, you mean the cost of these edges in the optimal solution, right?

l212 "...must be a pivot chosen among non-clustered...": Couldn't it also be because it's counter increased to $k$? It doesn't seem to matter for the argument?

Equation above l217: The first inequality is an equality, right? And $v\in U_{n+1}$ should be  $v\notin U_{n+1}$?

**Limitations:**

No limitations.

---

> ### Author Rebuttal · Authors · 2023-08-09
>
> 1. The algorithm by [CKLPU'22] uses $n \log(n)/\epsilon$ words of memory.
>
> 1. The memory usage is O(kn). It is linear in $n$ if $k$ is fixed.
>
> 2. __l131: I think the runtime bound can be improved to ...__
> Thank you! You are right.
>
> 3. That's correct.
>
> 4. You are right.
>
> 5. __When you say the optimal cost of the edges settled by PIVOT, you mean the cost of these edges in the optimal solution, right?__ That's correct. We will revise the wording to make it clear.
>
> 6. __l212 "...must be a pivot chosen among non-clustered...": Couldn't it also be because it's counter increased to $k$? It doesn't seem to matter for the argument?__
> We will more carefully explain what happens when the counter is increased to $k$. This case should be handled on l214, because in order for the counter $K_t(v)$ to get equal to $k$, it first needs to increase by 1.
>
> 7. That's correct. Thank you!

---

### Official Review · Reviewer_aXJT · 2023-07-06

**Soundness:** 4 excellent
**Presentation:** 3 good
**Contribution:** 4 excellent
**Rating:** 8
**Confidence:** 4

**Summary:**

This paper demonstrates that the celebrated Pivot algorithm for (unweighted, complete) correlation clustering (with the min disagreements objective) can be adapted to the single-pass, semi-streaming setting, to yield a (3+\eps)-approximation in expectation using O(n/eps) words of memory. (Note that the original Pivot algorithm gives a 3-approximation in expectation.) This improves on previous works that show a (3+\eps)-approximation using O(n\log n) words of memory, and a 5-approximation using O(n) words of memory. In the single-pass, semi-streaming setting, edges and their labels arrive online adversarially and the stream can only be read once. It is assumed that only positive edges arrive and the remaining edges in the complete graph are implicitly negative.

The Pivot algorithm chooses a random permutation of the vertices, visits each vertex in this order, and if a vertex is unclustered, makes that vertex a "pivot." The pivot grabs its unclustered positive neighbors from the remaining sequence and forms a cluster. The algorithm proposed in this paper tweaks the Pivot algorithm by again choosing a random permutation of the vertices, but for each vertex it only retains positive edges to the top-k ranked neighbors in this permutation. It then runs Pivot on this subgraph, using the chosen random permutation. The analysis rests on controlling the cost of (positive) edges cut by special vertices called "singleton vertices." These are vertices u whose top-k positive neighbors all come before u, but none are chosen as pivots, so u ends up in a singleton cluster. (Controlling the cost of other edges is done by reusing the analysis of Pivot, since these edges are cut by the pivot vertices.) The crux of the argument controlling the cost of singleton vertices is to show that a certain potential function is always positive in expectation, which in turn comes from showing it is a submartingale.

**Strengths:**

- The algorithm is a pleasingly simple variant of the Pivot algorithm. It lends insight into just how much "wiggle room" there is in the original Pivot algorithm, as it might a priori seem surprising that we only need retain the top-k ranked neighbors of each vertex.

- The analysis (particularly the main lemma, which designs a novel potential function and shows it is a submartingale) is quite elegant. Moreover, it is substantially simpler than analyses in the aforementioned previous works on the semi-streaming setting.

- This paper contributes to the growing literature on how the Pivot algorithm can be adapted to other models of computation (in addition to streaming, parallel models such as MapReduce). While the result itself is not a significant improvement over known results, and the algorithm is of comparable simplicity to previous ones, it provides a much cleaner analysis.

**Weaknesses:**

While the paper is overall clear and well-written, it would be useful to give a bit more background comparing the proposed algorithm to those in prior work in the semi-streaming setting.

Some typos / omissions in the writing:
- U_t on pg. 6 is seemingly meant as the complement of how it is defined on pg. 4
- D^+(v) = |N(v)| on pg. 5
- It would be good to add a line stating formally what it means to "cut" at an edge (although it is clear from context)

And a suggestion: The use of the term singleton is a bit confusing, since, as the authors mention, not all singleton clusters correspond to "singletons" as defined in the Pivot selection and clustering section of Figure 1. It may be worth just using a distinct term altogether for such vertices.

**Questions:**

n/a

---

> ### Author Rebuttal · Authors · 2023-08-09
>
> Thank you for your suggestions! We will revise the introduction and include more background on the prior work. We will also use a different name for "singletons". Finally, we will fix the _typos/omissions_.

---

> > ### Comment · Reviewer_aXJT · 2023-08-18
> >
> > I thank the authors for their responses. While it is true that the actual result is perhaps an incremental improvement, I like the analysis a lot and through it the insight into the Pivot algorithm gained. I maintain my evaluation.

---

### Official Review · Reviewer_mH8W · 2023-07-17

**Soundness:** 4 excellent
**Presentation:** 4 excellent
**Contribution:** 3 good
**Rating:** 7
**Confidence:** 5

**Summary:**

This paper provides a streaming algorithm for correlation clustering on complete graphs that uses O(n/eps) space and has an approximation factor of 3+eps. The algorithm is based on the combinatorial pivot algorithm.

The prior results in the streaming setting give 5 approx in the better space of O(n); and another result with 3+eps approx in space O(n log n). They are also based on the combinatorial Pivot algorithm.

The best offline approximation for the problem achieves a factor below 2, but the best approximation factor based on the combinatorial pivot algorithm achieves a factor of 3.

**Strengths:**

Matches the best offline approximation based on combinatorial Pivot algorithm

The algorithm is simple and thus practical to implement, and the proof is very clean.

**Weaknesses:**

They provide a code in the supplementary material but no experimental results comparing this algorithm with the prior works.

**Questions:**

Minor comments:
Page 1, sings -> signs
In description of Algorithm 2, did you mean U_1 = V as opposed to U_1=U?

---

> ### Author Rebuttal · Authors · 2023-08-09
>
> You are right $U_1 = V$. Thank you!

---

### Official Review · Reviewer_tNUH · 2023-07-26

**Soundness:** 3 good
**Presentation:** 3 good
**Contribution:** 2 fair
**Rating:** 4
**Confidence:** 4

**Summary:**

This paper considers the problem of correlation clustering on a complete graph, where all the edges are assigned either "+" or "-". The goal is to come up with a clustering that maximizes agreement with the edge labels. This is a well studied problems in approximation algorithms. There is an elegant 3 approximation due to ACN called PIVOT: pick unclustered vertices in random order, create a cluster for each such vertex and all unclustered neighbors that have a + edge to it. The analysis is similarly simple and elegant.

This paper considers the semi-streaming model. where the edges arrive one at a time. It is not obvious how the ACN version of PIVOT works in this model, since we do not know all the neighbors of a vertex v when processing it. A recent paper due to Behnezhad et al gives a 5 approximation for a variant of PIVOT. Their algorithm  gives a 5 approximation in O(n) space, but its analysis is subtle and a little tedious.

This paper presents another variant of pivot which gives a 3 +eps approximation, again in O(n/eps) space. The analysis is arguably simpler (but to my taste, not as simple as the original PIVOT algorithm).  Previously, there was a 3 +eps algorithm, but that needed n log n space.

**Strengths:**

It is nice to have both a simple algorithm and a simple analysis for a very natural problem. This paper deserves to be published somewhere.

**Weaknesses:**

The contribution feels somewhat incremental.  We had a simple algorithm that gave a 5 approximation in O(n) space, as well as a 3 + eps approximation in O(n log n) space. This paper gives 3 + eps with a simple algorithm and analysis, and O(n/eps) space. In some sense, it gets the best of all worlds.

I am not entirely convinced that either then space reduction or factor improvement ranks as a important enough contribution at a top ML venue like neurips. For calibration, I dont think this would get into a top tier theory venue, but I'd be fine with accepting it to a next-tier conference. If some area expert thinks this is the case, I'd be open to changing my mind.


**Questions:**

When you say that the stream only consists of positive edges, it is worht pointing out that you are working on the complete graph and the remainder of the edges are assumed to be negative.

---

> ### Author Rebuttal · Authors · 2023-08-09
>
> __When you say that the stream only consists of positive edges, it is worth pointing out that you are working on the complete graph and the remainder of the edges are assumed to be negative.__
>
> Thank you! That's correct. We will clarify this point in the final version of this paper.

---

### Author Rebuttal · Authors · 2023-08-09

We thank the anonymous reviewers for their valuable comments. We are grateful to them for carefully reading our paper, giving us helpful suggestions, and providing a list of typos. We will address all the feedback we received in the final version of this paper and, of course, fix the typos and add necessary clarifications.

We believe that our paper offers significant theoretical and practical advancements in the study of the Correlation Clustering problem. The contribution of our work is twofold: (1) Our algorithm demonstrates improved memory usage compared to both streaming and non-streaming algorithms previously introduced. (2) It is significantly simpler than other Correlation Clustering algorithms, except for the original Pivot algorithm. This simplicity makes our algorithm easy to implement. Our reference implementation is concise, consisting of fewer than 50 lines of C++ code. Moreover, our proof is just 3-4 pages long, which makes it accessible and comprehensible. The reason why we highlight the simplicity of our algorithm is that, in practice, people prefer to use simple algorithms whenever possible. There are many known sophisticated algorithms for Correlation Clustering, but, in practice, the algorithm of choice is, due to its simplicity, the original Pivot algorithm. In the case of streaming algorithms, our algorithm is both better and substantially simpler than other algorithms.

Our algorithm can also be used in the non-streaming setting when the data is stored in a file. In this case, the advantage of our algorithm is that it uses much less memory than the standard Pivot. We conducted experiments with two synthetic data sets. The plot (provided in our response as a separate PDF file) shows that the performance of our variant of Pivot is essentially the same as the performance of the original Pivot for $k \geq 4$. These data sets are based on the K-nearest neighbor (K-nn) graphs for points generated through a Gaussian mixture model. We normalized the cost such that the standard Pivot algorithm's cost is set at 1. After running our Pivot variant 100 times on each graph, we averaged the costs. Please, note that k denotes the parameter of our algorithm, while K denotes the number of nearest neighbors in the K-nn graph.

---

> ### Comment · Reviewer_NEv2 · 2023-08-14
> **Space usage**
>
> Thank you for providing additional details. Have you also measured the space usage of the baseline and your implementation?

---

> > ### Author Response · Authors · 2023-08-14
> >
> > We compared our algorithm with k = 4 and the original Pivot algorithm. The maximum memory usage slightly depends on the platform. On Linux (clang++ or g++ compiler), we got 17 MB (k = 4) vs. 38 MB  (original pivot). On Windows (Microsoft Visual C++), we got 17 MB (k = 4) vs 46 MB (original pivot). The number of edges stored in memory is 776K by our algorithm (k = 4) vs 5,200K by the original pivot.

---

> > > ### Comment · Reviewer_NEv2 · 2023-08-16
> > >
> > > Thank you for providing more information. I gave the paper one more thought and decided to stand by my earlier recommendation. While I very much like the algorithm, the theoretical result alone does not feel "big" enough for me to meet the NeurIPS bar. Adding empirical evaluation would have increased my score, but unfortunately it was not provided in the initial submission.

---

### Decision · Program_Chairs · 2023-09-21

**Decision:**

Accept (poster)

**Comment:**

The reviewers are split on this submission with some strongly supportive of the paper and some suggesting that the results are largely incremental.

The paper gives a new algorithm for correlation clustering in the low memory setting. The algorithm is a variant of the classical pivot algorithm with a simple and elegant analysis. The theoretical improvement over recent results are minor but the bounds seem to be as good as they can be. The original paper contained no experiments and thus it is hard to ascertain if the simplicity of the algorithm indeed translates into usefulness in practice. The rebuttal included some limited experiments that are more convincing and would strengthen the paper significantly if it is taken into account.